

# Going rogue: what scientists can learn about Twitter communication from "alt" government accounts

Matthew J. Wilson[1] and Elizabeth K. Perkin[2]

[1] Freshwater Research Institute, Susquehanna University, Selinsgrove, PA, United States of America
[2] Native Fish Society, Oregon City, OR, United States of America

## ABSTRACT

The inauguration of President Trump in the United States led to the active restriction of science communication from federal agencies, resulting in the creation of many unofficial "alt" Twitter accounts to maintain communication. Alt accounts had many followers (*e.g.,* 15 accounts had > 100,000) and received a large amount of media attention, making them ideal for better understanding how differences in messaging can affect public engagement with science on microblogging platforms. We analyzed tweets produced by alt and corresponding official agency accounts to compare the two groups and determine if specific features of a tweet made them more likely to be retweeted or liked to help the average scientist potentially reach a broader audience on Twitter. We found adding links, images, hashtags, and mentions, as well as expressing angry and annoying sentiments all increased retweets and likes. Evidence-based terms such as "peer-review" had high retweet rates but linking directly to peer-reviewed publications decreased attention compared to popular science websites. Word choice and attention did not reflect official or alt account types, indicating topic is more important than source. The number of tweets generated and attention received by alt accounts has decreased since their creation, demonstrating the importance of timeliness in science communication on social media. Together our results show potential pathways for scientists to increase efficacy in Twitter communications.

## INTRODUCTION

Since its inception in 2006, the microblogging social media platform Twitter has dramatically changed the science communication landscape, allowing scientists to communicate directly with a broad audience as well as one another. The abilities to directly comment to politicians on science policy issues and potentially influence media coverage are particular perks of this method of communication (*Newman, 2016*). However, many scientists create tight networks across media platforms where they primarily communicate with other scientists (*Wilson et al., 2016*) which can particularly undermine the potential of social media platforms like Twitter to reach large audiences and implement real-world change (*Letierce et al., 2010*). For instance, in one survey of scientists, 47% of researchers

Corresponding author
Matthew J. Wilson,
wilsonmatt@susqu.edu

who used social media to share their findings had been contacted by other researchers, while only 28% had been contacted by members of the public as a result of this outreach (*Wilkinson & Weitkamp, 2013*). Conversely, when scientists are able to expand their network of followers it has the potential to not only increase their citation rate (*Peoples et al., 2016*; *Lamb, Gilbert & Ford, 2018*), but also readership among the media and general public (*Côté & Darling, 2018*), demonstrating the value of social media platforms when used effectively. Further, analyses of microblogging data have improved our understanding of how topics such as conservation awareness and media coverage (*Acerbi et al., 2020*), common *vs* Latin species name use (*Jarić et al., 2016*), and informal citizen science (*Daume & Galaz, 2016*) reach lay audiences.

Government agencies involved with science and policy also seek to communicate scientific information to a broad audience. However, following the 2017 inauguration of President Trump, the mandate of government agencies within the United States to communicate scientific findings was questioned by the executive branch of the government, including the prohibition of official accounts from tweeting climate-related information (*Volcovici & Huffstutter, 2017*). Employees within many of these U.S. agencies created "alt" or "rogue" (henceforth referred to as "alt") Twitter accounts as a way to continue to share information as well as criticize the previous presidential administration within the United States (*Davis, 2017*).

These alt Twitter accounts gained coverage across a wide variety of journalistic platforms, from blogs (*e.g.*, LiveScience: *Weisberger, 2017*) to cable news (*e.g.*, CNN: *Walker, 2017*) and developed large numbers of followers. Because these alt accounts had more followers than most individual scientists, we were interested in determining if the scientific community might gain valuable insight for better reaching and broadening their Twitter audience by studying the habits of alt and corresponding official Twitter accounts of US agencies. Specifically, we were interested in what topics receive the most attention in alt *versus* official accounts, if there is a large discrepancy between the hashtags and mentions between alt and official accounts, what types of keywords and links (*e.g.*, peer-reviewed papers or popular science websites) garner the most attention, and how attention to alt accounts changed over time. We limited our study to focus on agencies involved in conservation biology, as we anticipated that patterns in hashtags, links, and keywords would be more readily apparent by focusing on one area of science.

## MATERIALS AND METHODS

### Data collection

We identified all official US federal Twitter accounts with an "alt" or "rogue" corollary handle that could be linked with a single federal agency and collected total followers, tweets, likes, and accounts followed for each account on 1 April 2017 by searching Twitter with the terms "rogue" and "alt" then identifying the official account that was the intended target of each alt account (Table S1). To identify additional unofficial accounts under other titles (*e.g.*, @BadHombreNPS) we scanned followers of the unofficial accounts previously identified with the assumption that alt accounts would form networks with each other.

We collected original tweets for alt accounts since their creation (generally 24 Jan 2017), and tweets between 1 Jan 2016 and1 April 2017 for official accounts, using the Java project GetOldTweets to avoid date constraints of the Twitter Application Programming Interface (*Henrique, 2016*). These tweets were collected two weeks after the final date posted (14 April 2017) to decrease the likelihood total retweets and likes of the most recent tweets were underestimated (popular tweets might be liked and retweeted for several days after posting; *Luo et al. 2015*). All tweets were also collected within a three hour period to limit differences in retweet and like numbers further, as new retweets or likes could be registered during collection. On 1 April 2017, there were 120 active alt accounts that could be linked directly to 71 official Twitter accounts of the US government (Table S1).

We selected only those official-alt account pairs in which the most prominent alt account had at least 50,000 followers on 1 April 2017. From these criteria we selected the official and top alt account pairs for National Parks Service (NPS), National Oceanic and Atmospheric Administration (NOAA), US Environmental Protection Agency (USEPA), US Department of Agriculture (USDA), US Forest Service (USFS), and US Fish and Wildlife Service (USFWS). We also included the official and alt accounts for Badlands National Park, as the alt account for this park had more followers (209,000) than the alt account for NPS (89,000). The primary alt account for NPS changed names on 29 Jan 2017. We included both handles (@NotAltWorld and @AltNatParkSer) and their corresponding tweets separately in analyses by account to avoid confounding results by the change in followers corresponding to this name change. With the inclusion of both NPS alt handles, the dataset included 15 handles and 9,688 tweets.

## Tweet emotion and features

To examine the potential effects of tweet emotion on attention (measured as retweets and likes) we assigned emotions to each tweet with the DepecheMood lexicon database, based on social news media with high lexical precision (*Staiano & Guerini, 2014*). This high-coverage database characterizes 37,000 English terms by seven possible emotions: afraid, amused, angry, annoyed, happy, inspired, and sad. Each term in the database is associated with all emotions on a 0–1 scale so that the combined value for all emotions conveyed by a single term is 1. To give overall emotions to tweets, we applied the normalized DepecheMood database to each tweet and used the mean value for all emotions across all terms in the tweet to represent overall emotion. We selected DepecheMood for the high coverage of terms and large corpus, as well as the ease of access and implementation for broad applicability. See *Giachanou & Crestani (2016)* for a detailed review of Twitter sentiment methods.

For this analysis, we focused on how tweet emotion and tweet features (URLs or photos attached, number of mentions, and number of hashtags) influenced the relative number of retweets and likes, while controlling for the effects of twitter handles and account type (official and alt) since absolute number of retweets can be influenced by number of followers. We used a Generalized Additive Mixed Modeling (GAMM) approach to accomplish this as we did not expect a linear relationship with predictors and human behaviour (*Wood, 2011*). We anticipated the alt movement could affect attention received

by official agency accounts. To take this into consideration for our GAMM we only used official tweets between the creation of the first alt account (24 Jan 2017) and 1 April 2017 as the "post-official" tweets, while "pre-official" tweets came from between 24 Jan 2016 and 1 April 2016 to avoid possible effects of the election campaign and differences in season-specific tweets (*e.g.*, potential for more tweets about winter/spring than summer/fall in January-March). Our global model included all DepecheMood emotions (except for the "don't care" category) and tweet features with twitter handles and account type (pre-official, post-official, and alt) as random effects. To estimate the best possible model, we used a model-averaging approach where all possible combinations of fixed effects are tested and models with $\Delta AICc \leq 2$ from the single best model are weighted and averaged together (*Burnham & Anderson, 2002*).

## Science terms and retweets

To determine if specific terms increased the likelihood of attention, we created a list of science- and evidence-based terms used in tweets, by splitting all tweets into individual terms (words within tweet text and hashtags) to compare with the associated number of retweets and averaged retweets by the associated term frequency. We manually identified all unambiguous science and evidence related terms used in tweets (*e.g.*, "vulture" and "rat", which were often used to vilify people and/or other agencies, were excluded) and merged singular and plural versions of terms into a single value. We excluded agency related terms (*e.g.*, #usda or #epa) from the final matrix. This left a term by count and average retweet matrix of 496 terms (Table S2). For comparisons of this term matrix by account type (pre, post, alt) and agency we used Canonical Correspondence Analysis (CCA) to deal with the unbalanced sample size across groups (*e.g.*, high numbers of tweets from alt compared to official accounts). Agency and account type were tested for significant effects on overall science term use and term retweets *via* ANOVA-like permutations (*ter Braak & Verdonschot, 1995*; *Legendre, Oksanen & ter Braak, 2011*) with the vegan package in R (*Oksanen et al., 2017*; *R Core Team, 2016*). While not used for analysis, we categorized the science term matrix further (*e.g.*, by "ecosystem", "evidence", "organism") to identify which terms within categories received the most attention and how categories were associated with CCA results. To be conservative, terms ambiguously within multiple categories were left uncategorized (Table S2).

## Tweet links

We investigated the effect of linking to science-specific websites in tweets compared to other link types and divided these into direct peer-review links to publications and links to popular science websites. Tiny URLs attached to tweets were decoded by the stand-alone "decode_short_url" R function (*Breyal, 2012*) and manually verified. The majority of peer-review links were tweeted by @AltUSEPA (15 of 22 tweets), therefore we restricted our analysis of lay and peer-reviewed links to this account. We compared retweets and likes between groups with a Kruskal-Wallis ranked ANOVA for multiple comparisons corrected for ties with the PMCMR package (*Pohlert, 2014*; *Conover & Iman, 1979*).

### Time series

To test the significance of timeliness in tweet attention, we performed multiple linear regression of total retweets, retweets per tweet, total likes, and likes per tweet against the number of days after creation of the first alt account, grouped by alt and official accounts. We excluded the date the first alt account was created (24 Jan 2017) to remove the skewing effects (*i.e.,* very high attention on Twitter) of tweets made by all accounts on that date. All reported r-squared values are adjusted r-squared.

## RESULTS

### Tweet emotion and features

Most emotions did not significantly affect retweets or likes. However, tweets that conveyed anger significantly increased both retweets and likes, annoying language significantly increased likes, and inspired language significantly decreased both retweets and likes. Angry language also had the strongest effect on number of retweets and likes. Higher numbers of hashtags and mentions and the attachment of images or links to tweets all significantly increased both retweets and likes. These effects were smaller than significant emotions and also had smaller confidence intervals (*i.e.,* more consistent effects) than emotions. There were no significant effects of account type (official-pre, official-post, or alt; Table 1).

### Science Terms and Retweets

Canonical Correspondence Analysis results showed a significant effect of US agency on science-related word and hashtag use in tweets (adjusted $R^2 = 0.14$, $p < 0.001$). While there was some visible separation of word choice by account type (pre, post, and alt), this relationship was not significant (Fig. 1A). This pattern was primarily the result of higher use of terms related to chemistry, evidence, fossil fuel, pollution, and climate by alt accounts, as well as NOAA and USEPA accounts. When looking at the results by agency, differences were most visible along Axis 2, with tweets from the USFWS accounts containing more organisms (particularly birds, reptiles, and amphibians) while activities (*e.g.,* camping, fishing, hiking) were most closely related to tweets from USFS accounts (Fig. 1B). Neither grouping accounts by agency nor by type had a significant effect on retweets of science terms. The effect of agency on retweets by word were negligible ($R^2 < 0.05$). Words and hashtags that received the most attention for climate change-related terms were related to current fossil fuel extraction (*e.g.,* "keystone" and "spill"), "peer-review" and "conferences" were the most retweeted evidence terms, flowers and large mammals were the most retweeted organism groups, and prairies and forests were the top retweeted ecosystems (Table 2).

### Tweet links

For tweets by @altUSEPA, the addition of website links significantly increased both retweets ($99.6 \pm 7.1$, $p < 0.0001$, $n = 522$) and likes ($173.4 \pm 12.9$, $p < 0.0001$, $n = 522$) over tweets without links ($19.9 \pm 2.4$ retweets/tweet and $73.0 \pm 8.6$ likes/tweet, $n = 1374$). This increase in attention was further amplified by the use of links to popular science websites ($127.4 \pm 32.0$ retweets/tweet and $232.0 \pm 59.3$ likes/tweet; $n = 52$ and $p < 0.0001$ for both

**Table 1  GAMM results from model averaging for mood, tweet features, and accounts.** Significance levels are <0.001***, <0.01**, <0.05*, <0.10: total number of models incorporated is given by s; $R^2$ is adjusted $R^2$; both averaged models had a weight of 1; and the intercept estimate is the effect of mean values for moods, hashtags, and mentions without links or images (binary factors).

| | Retweets $R^2 = 0.59$ s = 8 | | Likes $R^2 = 0.66$ s = 5 | |
|---|---|---|---|---|
| | Estimate | Confidence Interval | Estimate | Confidence Interval |
| (Intercept) | **4.00** | (3.24 to 4.75)*** | **4.75** | (3.98 to 5.52)*** |
| **Mood (Fixed Effects)** | | | | |
| afraid | 0.33 | (−0.27 to 0.93) | −0.07 | (−0.33 to 0.20) |
| amused | −0.22 | (−0.71 to 0.27) | 0.04 | (−0.18 to 0.26) |
| angry | **4.45** | (3.77 to 5.13)*** | **4.19** | (3.61 to 4.76)*** |
| annoyed | 0.45 | (−0.36 to 1.27) | **2.14** | (1.41 to 2.88)** |
| happy | 1.19 | (0.18 to 2.20) | 0.15 | (−0.28 to 0.58) |
| inspired | **−2.98** | (−3.69 to −2.27)*** | **−1.35** | (−1.95 to −0.75)* |
| sad | −0.09 | (−0.45 to 0.27) | −0.12 | (−0.49 to 0.25) |
| **Tweet Features (Fixed Effects)** | | | | |
| Hashtags | **0.17** | (0.14 to 0.20)*** | **0.15** | (0.12 to 0.17)*** |
| Mentions | **0.11** | (0.06 to 0.15)* | **0.12** | (0.07 to 0.16)** |
| Links | **0.82** | (0.77 to 0.86)*** | **0.31** | (0.27 to 0.35)*** |
| Images | **0.75** | (0.68 to 0.82)*** | **0.44** | (0.38 to 0.51)*** |
| **Account Type (Random Effects)** | | | | |
| Alt Account | 0.69 | (0.04 to 1.35) | 0.77 | (0 to 1.53) |
| Official Post | 0.21 | (−0.34 to 0.77) | 0.42 | (−0.25 to 1.09) |
| Official Pre | −0.90 | (−1.46 to −0.35) | −1.18 | (−1.86 to −0.51). |
| **Accounts (Random Effects)** | | | | |
| BadlandsNPS | 0.72 | (0.25 to 1.18) | 0.60 | (0.13 to 1.07) |
| BadHombreNPS | **1.36** | (0.82 to 1.90)* | **1.47** | (0.94 to 2.00)** |
| EPA | **−1.20** | (−1.66 to −0.74)** | **−1.36** | (−1.83 to −0.90)** |
| altUSEPA | **3.41** | (2.94 to 3.89)*** | **3.39** | (2.92 to 3.87)*** |
| forestservice | **−1.85** | (−2.31 to −1.38)*** | **−1.68** | (−2.14 to −1.22)*** |
| AltForestServ | **−1.15** | (−1.65 to −0.65)* | −0.93 | (−1.43 to −0.43) . |
| NatlParkService | **1.57** | (1.09 to 2.05)** | **1.08** | (0.60 to 1.56)* |
| AltNatParkSer | −0.95 | (−1.44 to −0.45) . | **−1.08** | (−1.57 to −0.58)* |
| NotAltWorld | −0.24 | (−0.76 to 0.28) | 0.07 | (−0.44 to 0.59) |
| NOAA | 0.38 | (−0.10 to 0.86) | 0.48 | (0 to 0.95) |
| altNOAA | −0.60 | (−1.10 to −0.09) | −0.70 | (−1.21 to −0.20) |
| USDA | 0.02 | (−0.45 to 0.48) | −0.04 | (−0.50 to 0.42) |
| altusda | −0.29 | (−0.80 to 0.23) | −0.33 | (−0.84 to 0.18) |
| USFWS | **−1.32** | (−1.79 to −0.86)** | **−1.28** | (−1.74 to −0.81)** |
| AltUSFWS | 0.14 | (−0.35 to 0.64) | 0.32 | (−0.18 to 0.81) |

retweets and likes) compared to tweets without links. However, linking to peer-reviewed publications rather than popular science significantly decreased the number of retweets ($44.2 \pm 18.4$, $p = 0.02$, $n = 15$) and likes ($78.1 \pm 35.7$, $p = 0.032$, $n = 15$) compared to other websites and was not significantly different from tweets without links.
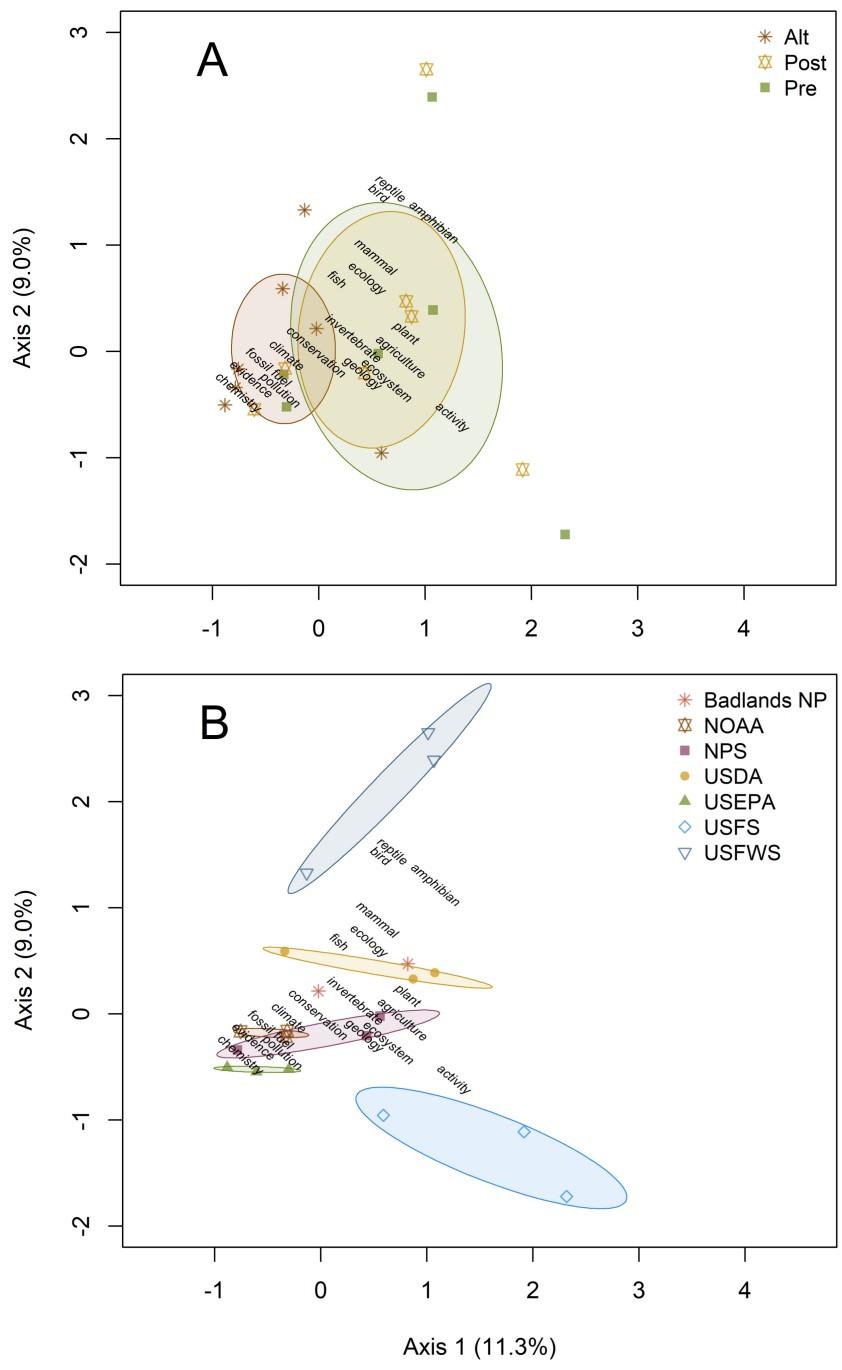

**Figure 1  Canonical Correspondence Analyses comparing science related term and hashtag use by accounts.** Percent variance explained by axes is given in parentheses by (A) account type (alt, pre-official, post-official) and (B) agency. Ellipses represent 95% confidence intervals around the centroid for each group.

## Time series

Tweets from alt accounts showed a significant decrease in total retweets ($R^2 = 0.13$, $p = 0.002$), retweets per tweet ($R^2 = 0.16$, $p = 0.0005$), total likes ($R^2 = 0.12$, $p = 0.002$),

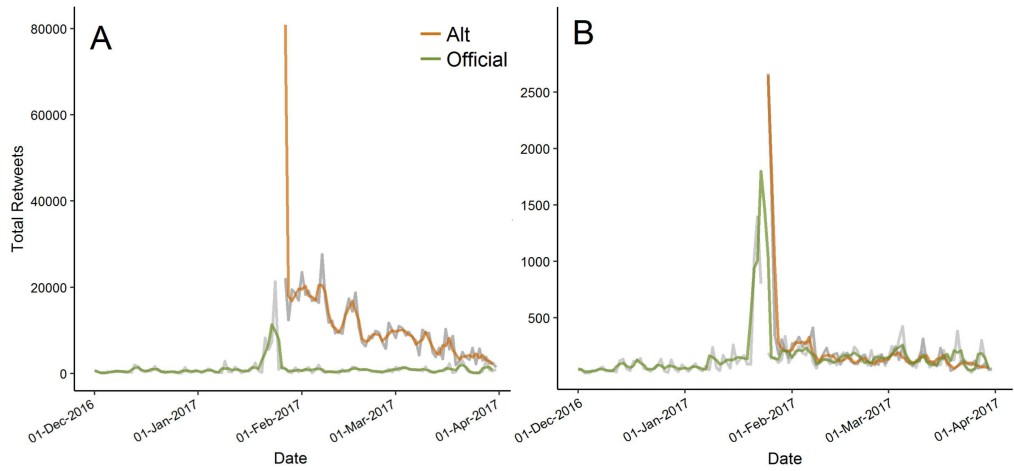

**Figure 2** **Total retweets summed across all accounts through time.** (A) retweets per tweet (as means across accounts); (B) comparing official and alt accounts between 1 Dec 2016 and 1 Apr 2017 (alt accounts began on 24 Jan 2017). Grey lines are daily values and colored lines are a three-day moving average for clarity. The $y$-axis is truncated at the moving average maximum for clarity.

and likes per tweet ($R^2 = 0.15$, $p = 0.0008$) between 25 Jan 2017 and 1 April 2017 (Fig. 2). None of the attention metrics tested for official accounts had significant temporal trends (all $R^2 < 0.03$).

## DISCUSSION

By closely examining tweets from both alt and official government accounts, we were able to discover underlying variables within the tweets that could help predict if a tweet would be liked or retweeted. We were surprised to see tweets categorized as "angry" were both significantly more likely to be retweeted and to be liked, as *Hutto, Yardi & Gilbert (2013)* found negative sentiment in tweets to have a negative effect on the number of followers accrued over time. One possible explanation for these opposing findings might be a general sentiment of anger felt by people who supported these government accounts at this point in time and may not represent the best tactic for most scientists to increase outreach. Exceptions might be when one wants to take advantage of a particularly emotionally charged event. However, leveraging emotion, especially politically-motivated anger, could actually result in these alt accounts reinforcing ideas that scientists are elitists and work only for Democrats, while further alienating Republicans (*Nisbet & Scheufele, 2009*). On the other hand, the finding that tweets containing hashtags, mentions, images, and links (images and links in particular) accumulate more retweets and likes, is a strategy that all scientists can easily incorporate into their tweeting practices in order to increase their reach and mirrors results from US health agencies (*Bhattacharya, Srinivasan & Polgreen, 2014*). This finding could also be valuable for scientists in gaining trust *via* Twitter as tweets

Wilson and Perkin (2021), *PeerJ*, DOI 10.7717/peerj.12407

**Table 2  The twenty terms most likely to be retweeted by category.** Terms were ranked if they occurred in four or more tweets (median number of tweets for terms); RT/T refers to retweets per tweet and n refers to the number of tweets.

| | Ecosystem | | | Evidence | | | Organism | | | Climate Change | | |
|---|---|---|---|---|---|---|---|---|---|---|---|---|
| Rank | Term | RT/T | n | Term | RT/T | n | Term | RT/T | n | Term | RT/T | n |
| 1 | prairie | 1248 | 13 | peer-reviewed | 2804 | 7 | violets | 904 | 4 | keystone | 5326 | 4 |
| 2 | forest | 940 | 148 | conference | 2579 | 4 | mammal | 665 | 10 | spill | 1671 | 6 |
| 3 | plains | 523 | 6 | knowledge | 2486 | 10 | rose | 621 | 9 | diesel | 1304 | 6 |
| 4 | streams | 440 | 12 | verify | 2156 | 6 | bison | 613 | 7 | atmosphere | 1007 | 18 |
| 5 | ecosystem | 343 | 8 | #antiscience | 2044 | 6 | wolf | 460 | 12 | #standingrock | 960 | 5 |
| 6 | reef | 320 | 8 | denial | 1968 | 20 | otter | 452 | 4 | oil | 884 | 30 |
| 7 | sea | 234 | 25 | censorship | 1902 | 10 | coral | 441 | 5 | #allpipesleak | 880 | 4 |
| 8 | coast | 232 | 6 | evidence | 1701 | 18 | fish | 348 | 11 | climate | 813 | 226 |
| 9 | arctic | 218 | 9 | study | 1145 | 15 | animal | 301 | 87 | #nodapl | 795 | 21 |
| 10 | ocean | 177 | 29 | scientific | 1133 | 50 | chicken | 286 | 8 | pipeline | 725 | 26 |
| 11 | #worldwetlandsday | 163 | 6 | facts | 897 | 95 | shark | 266 | 5 | #nokxl | 697 | 4 |
| 12 | lake | 127 | 12 | investigate | 715 | 6 | eagle | 233 | 16 | dioxide | 674 | 15 |
| 13 | beach | 125 | 7 | accurate | 547 | 7 | owl | 233 | 15 | carbon | 578 | 23 |
| 14 | woods | 109 | 9 | data | 338 | 62 | chick | 216 | 7 | #climatechange | 478 | 66 |
| 15 | wetland | 105 | 13 | think | 271 | 113 | seal | 194 | 10 | #dapl | 438 | 8 |
| 16 | river | 100 | 33 | analysis | 253 | 5 | eaglet | 186 | 4 | fracking | 303 | 11 |
| 17 | island | 95 | 8 | published | 223 | 4 | tree | 172 | 48 | drilling | 296 | 9 |
| 18 | desert | 81 | 4 | explain | 180 | 21 | #dceaglecam | 171 | 7 | temperature | 249 | 7 |
| 19 | cuyahoga | 71 | 4 | proof | 156 | 6 | fox | 168 | 21 | $CO_2$ | 209 | 30 |
| 20 | #forestfriday | 62 | 4 | #climatefacts | 149 | 41 | albatross | 166 | 5 | #climatechangeisreal | 196 | 16 |

containing links have been viewed as more credible, regardless of the linked website (*Aigner et al., 2017*).

The terms used in tweets that received the most retweets were generally evidence- or climate-based. Words including "peer reviewed," "knowledge," and "verify" were associated with extremely high levels of retweets per tweet. This suggests researchers should feel comfortable using common science-based terms and that a large audience supports these terms. While ecosystem- and organism-specific terms did not receive as many retweets as those tweets containing evidence-based or climate-specific terms, there were still some important trends to emerge out of those categories, with ecosystems that people frequently come into contact with (*e.g.*, prairie, forest, streams), and flowers and large mammals (*e.g.*, violets, bison, wolf) leading to higher numbers of retweets. Similar patterns emerged in previous research examining which species listed under the Endangered Species Act were most tweeted (*Roberge, 2014*). While there were themes in the most common categories tweeted by alt accounts that might be expected (*e.g.*, fossil fuel, evidence, and climate), our CCA analysis revealed that word choice in tweets was strongly determined by agency (*e.g.*, NOAA, USFS), but not by type (pre, post, alt), suggesting alt accounts might really be run by agency personnel, as claimed.

Given the number of retweets of tweets containing evidence-based terms, particularly those tweets containing the phrase "peer reviewed," it surprised us that tweets containing links to actual peer reviewed papers were less likely to be retweeted or liked than those with other links. This suggests that while it can be tempting to share and promote peer reviewed literature on Twitter, this is not the best way to reach a broader audience. Rather, scientists interested in reaching a broader audience would be well advised to reach out to traditional media (*Nisbet & Scheufele, 2009*) and then share links to those "interpreted" stories. Indeed, we found tweets containing links to popular science articles significantly increased the likelihood that a tweet would be liked or retweeted. Interestingly, *Holmberg & Thelwall (2014)* found that scientists rarely use links when discussing a topic with one another on Twitter, but based on our findings, researchers may want to change this habit if they would like to expand their audience.

Perhaps one of our most important results was finding how timeliness plays a role in public interest in a topic. The number of retweets per tweet for both official and alt government accounts peaked soon after the inauguration of President Trump, when public interest in these government agencies was extremely high. However, very soon after, interest in all accounts returned to background levels. As a result, it might be important for scientists to take advantage of large, well-publicized events to tweet important information in order to reach a broad, interested, and engaged audience. When analyzing the popularity of several scientific terms on Twitter, *Uren & Dadzie (2015)* found similar results, with very high levels of interest in "curiosity" in the few days surrounding the successful landing of the Mars Curiosity rover and on the anniversary of the landing a year later, but with very low levels of interest beyond that time-frame.

At the time of our study, these alt and official government accounts presented a unique opportunity to discover elements of scientific tweets that influence how often they are liked or shared. However, in the years since we developed our hypotheses, more scientists

have gained large followings on Twitter that could allow for a similar analysis of individual scientists. Future work might examine how people respond to different emotions and links (*i.e.,* to peer-reviewed or general media outlets) in tweets about COVID-19. Examining how racism effects interactions with scientific tweets is another potential area of research. For instance, people may react differently when anger about an environmental injustice is tweeted by a scientist who identifies as #BlackinSTEM *versus* a white-presenting scientist.

## CONCLUSIONS

Twitter can be an important part of an effective science outreach and communication strategy (*Cooke et al., 2017*; *Parsons et al., 2013*), but it can be difficult to reach people outside well-developed scientific networks. Studying the strategies of alt government accounts allowed us to learn what emotions and tweet characteristics were associated with higher levels of likes and retweets. Alt accounts were more popular than official accounts immediately after their creation. However, the attention alt accounts received decreased over time, demonstrating the importance of timeliness in science communication. Word choice and word attention did not reflect whether accounts were official or unofficial, indicating the topic was more important than the source. While angry and annoying language in tweets increased attention metrics, inspirational language decreased attention. The presence of links and photos as well as higher numbers of hashtags and mentions also increased attention, suggesting increasing efficacy in science communication on social media is possible without negative sentiments in tweets. Linking to popular science from tweets was also more popular and more effective than linking directly to peer review, reinforcing the value of lay abstracts and press releases in communicating scientific results.

## ACKNOWLEDGEMENTS

We thank federal US employees for maintaining scientific integrity, perseverance, defiance, and creativity in continued science communication with the public. We particularly thank those anonymous individuals who began the alt movement on social media. This manuscript was improved by comments from AM Chará-Serna, BW Kielstra, TK Muenz, M Takahashi, and three anonymous reviewers.

### Funding

This work was supported by Susquehanna University. The funders had no role in study design, data collection and analysis, decision to publish, or preparation of the manuscript.

### Grant Disclosures

The following grant information was disclosed by the authors:
Susquehanna University.

### Competing Interests

The authors declare there are no competing interests.

## Author Contributions

- Matthew J. Wilson conceived and designed the experiments, performed the experiments, analyzed the data, prepared figures and/or tables, authored or reviewed drafts of the paper, and approved the final draft.
- Elizabeth K. Perkin conceived and designed the experiments, prepared figures and/or tables, authored or reviewed drafts of the paper, and approved the final draft.

## Data Availability

The data and code are available from GitHub:

https://github.com/wilson-matt/MicrobloggingAnalysis.

## Supplemental Information

Supplemental information for this article can be found online at http://dx.doi.org/10.7717/peerj.12407#supplemental-information.

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
