# Peer review of "Going rogue: what scientists can learn about Twitter communication from “alt” government accounts"

_PeerJ, doi:10.7717/peerj.12407_

## Round 0.1 · original submission · Major Revisions

The three reviewers I have consulted agree about the novelty and general interest of the results you obtained in the case-study application. They have raised, however, a number of questions regarding the statistical analysis, adequacy of your literature review, and (see especially reviewer #1) the claim of novelty for the methodology you have chosen to implement in the work. I believe the comments can be addressed but will require major revisions in the current version of the manuscript.

Reviewer 1 ·

Basic reporting

>> clear english

Fairly ok.

>> literature references and context

Poor.

>> article structure, figures, tables. data

Structure is okay, but not the content.

>> results to hypothesis

The main analysis done by authors are summarised in lines 117-124.

The second point on the statistical work to determine factors that drive attention is a valid one, and the authors have done this job reasonably well, except that they have both intercept and all levels of a categorical variable in their model.

The first point is baseless. They have compared nothing -- they have only used specific methods (already existing) for specific tasks. Next, I do not understand what is "novel" in testing that a certain non-parametric test can be "applied".

Experimental design

>> original primary research; within scope of journal

Yes, the topic is within scope of journal.

>> research question well defined; explains filling of knowledge gap ??

The authors claim that there are very few studies in microblogging analysis that have combined both qualitative text analysis with more quantitative network analyses. They claim that mixed-method approaches have become common in health care research -- it seems that authors are inspired by the usage of mixed-method in some health related papers.

The authors say that quantitative microblogging analyses have followed two main forms: network analyses and unsupervised algorithms modeling, and continue to claim that the usage of such methods outside computer science fields has been limited. I do not understand how the authors are claiming such things -- may be they have read several papers in all fields.

In lines 73-84, the authors describe their knowledge of what methods have been applied to microblogging studies. In my opinion, the authors have very limited knowledge on what methods have been applied to understand various issues in microblogs like Twitter. A wide range of simple to sophisticated methods in areas of statistics, machine learning, networks, time series, etc. have been used already. I do not understand why the authors say that "the methods employed do not extend beyond those described above" in line 82-84.

In lines 99-100, the authors say that "there have been multiple studies to confirm automated sentiment coding as an effective and time-saving proxy for manual coding", but they "have been unable to find any mention of their use singly or in conjunction for social media text and interest analyses". This is a surprising statement -- there are tons of papers out there on Twitter data, which have used sentiment analysis -- on large data sets like Twitter nobody does manual coding of sentiments!!

So, upto this point, it clearly shows that the authors are trying to fill a gap of "applying some menthods" to microblogging analysis. They think that they are the first ones to apply such methods -- and its not true. So they are not filling any knowledge gap which they claim in the title of the paper as "quantitative methods". First, they have not developed any method. Second, methods which they are "applying" to their dataset can not be claimed as a "new method".

Next, in line 108, they write "To test methods against a real dataset...". They have not developed any method, I do not understand what they are testing. In line 118, they say they are "testing the applicability of non-parametric test". Any method, in general, can be applied to contexts that lie within the assumption of the method. I don't understand what the authors are testing.

In line 120-121, the authors say that they are interested to "determine what elements of a tweet make it more likely to be shared and accrue likes". This is the only statistical thing with some hypothesis that the authors are really doing in this paper. Everything else they have written about "quantitative methods" is baseless, naive, and shows that they have no idea of existing literature.

Again, in line 125, the authors mention "In testing these methods...". Frankly there is no method they are testing -- they are only applying existing methods to a microblogging context, which they think is very novel.


>> data

The data collected is of good quality.

>> rigorous investigation, ethical standard

The authors have used GAMM to investigate the effect of tweet sentiment and various features of tweet on attention (retweet, likes).

There are 7 types of sentiments in their analysis -- afraid, angry, etc. In Table 1, they have used all 7 types and they also have the intercept term! Usually when a variable has N categories, N-1 categories are included in the model if the model has an intercept. This is to avoid multicolinearity.

>> methods described with sufficient detail & info to replicate

In line 279, the authirs say "GAMM was an effective method for identifying underlying variables that could predict if a tweet would be liked or retweeted". The only method they have used is GAMM, and now they are saying it is an effective method -- to what other methods are they comparing GAMM to?

Validity of the findings

>> impact and novelty

The only results which are interesting are (1) tweets categorized as “angry” were both significantly more likely to be retweeted and to be liked, (2) words including “peer reviewed,” “knowledge,” and “verify” were associated with extremely high levels of retweets per tweet, and (3) tweets containing links to actual peer-reviewed papers were less likely to be retweeted or liked than those with other links.

The authors have expressed their intuition for such observation in the discussion section.

In my opinion, these are the "real results" from the authors in this paper. To me, these results are surprising and novel. The entire paper should have been written around these results. I do not know why the authors emphasized a lot on trying to write about their novelty on "quantitative methods" -- there is none in their work.

>> underlying data have been provided; replicable

With the list of Twitter accounts provided in supplementary, a researcher can ideally collect that data from Twitter and replicate the authors' study.

>> well stated conclusions; linked well to research question; limited to supporting results

The title of the paper is "Quantitative methods and perspectives..". Frankly, I do not see what new methods have been proposed in the paper. The authors have only used existing methods to analyse a particular hypothesis.

The abstract is poorly written. The last line of abstract -- "Overall, our results highlight analytical methods to quantify audience behaviour in microblogging platforms, informing possible pathways for science communications" -- is completely misleading. The authors have only used some methods to analyse their hypothesis. This does not mean that they are quantifying or measuring behaviour -- they are only investigating the nature of a particular behaviour.

Reviewer 2 ·

Basic reporting

The article is written with a professional use of English (but I am not a native speaker). References about previous studies justifying the goal of the study, methods used are included. Maybe, authors could try to cite more recent papers because the number of papers published analysing Twitter data has increased in last times.
The structure of the paper follows the standard indicated by the journal. It contains an introduction section, material & methods, results, discussion and conclusions. I recommend review the structure of material & methods because it contains 5 subsections and, for example, subsection 2.5 is summarized in 6 lines (maybe it is not necessary too many subsections). In addition, the results section should be reviewed because, for example, when describing Table 1 I think that some parts are missing.
Figures are relevant to check results, nevertheless, I would try to improve the resolution because they are quite difficult to analyze due to the size of the image (specifically, Figure 1 with CCA).

Experimental design

The paper is within the Aims and Scope of the journal.
The abstract of the paper states two main objectives. First, to compare communication strategies and test the applicability of different tests, methods. Second, to analyze what elements of a tweet gain attention. In my opinion, after reading the paper, the general idea that the reader gets is how to use different methods to analyze data from microblogging. I mean, the second objective of the paper is less present until the section of results. I think that during the introduction this part should be emphasized a little more because it seems that the important/novelty is the use of CCA, GAMM, etc and not where alt accounts matter, whether the content of the tweet matters, etc. As I said, It is not until the results or even discussion section, when reader notes the importance of sentiments, etc (and is one of the goals of the paper)
Regarding the material & methods section, in my opinion, authors describe in a good way (with full of details), the gathering of data, cleaning process, etc.

Validity of the findings

I think that this study presents novelty results and give ideas for future analyses using Twitter data, for example, new statistical analyses that can be employed from other fields.
I would recommend to author to specify the main limitations of their analysis. In addition, I would like to know if these results could be generalized to alt and official accounts but considering other issues and not the environmental issues? I would also like to know what is the future direction?

Additional comments

Thank you for providing the opportunity to review the manuscript. I think that this paper is interesting and novel analyzing microblogging data paying special attention to methods but also the phenomenon of alt vs. official accounts.

Reviewer 3 ·

Basic reporting

The manuscript is generally readable and well written. There's a bit of awkward language here and there (266-269 runs on and uses "from" in an odd manner for example), but nothing which confuses.

1. The term sentiment is used in place of emotion to reference the DepecheMood attributes. In their original manuscript (Staiano and Guerini 2014) the authors of the lexicon deliberately distinguish their emotional lexicon from sentiment based lexicons.

2. I do not agree with the statement that Staiano and Guerini 2014 "confirm automated sentiment coding as an effective and time saving proxy for manual coding". They concluded "showing significant improvements over state-of-the-art unsupervised approaches" not equivalence to manual coding. This is, admittedly, a fairly minor nitpick but the ANGER affective dimension was firmly in the middle of the pack for matching human annotation.

Experimental design

1. This manuscript does not seem to fit into the scope of the journal. The focus of the manuscript is the application of statistical approaches to the analysis of data regarding microblogging about science topics. The manuscript explicitly states one of the goals is to analyze "communication strategies" (117-118) and that another is to test the applicability of "non-parametric tests, multivariate ordinations ... " (118-119) I would recommend targeting a journal with a focus on communication, science communication, or statistical analysis.

2. The DepecheMood lexicon annotates each term with eight "affective dimensions" not seven. The eighth "don't care" dimension is not mentioned in the manuscript.

3. No github link or version number is provided for the DepecheMood lexicon. There are currently two versions available (1.0 and 2.0 aka DM++). In order to replicate the work please specify which version was used.

4. It's not clear how "all official U.S. federal Twitter accounts with an "alt" or "rogue" corollary ..." (132 - 134). were identified. What strategy was used to identify official accounts and, more problematically, "alt" versions? Please describe how the search was accomplished.

5. How the data was collected is unclear (132-144). This would be less problematic if the collected data was provided in the supplementary material, but as it isn't more clarity would be appreciated. The tool used to gather the data could also be more clearly specified as "Get Old Tweets" rather than "a Java project" as the latter is more ambiguous.

Validity of the findings

1. A number of the conclusions seem to be overstated based on the relatively small sample size and lack of context. For example, the conclusion that "researchers should feel comfortable using common science-based terms ..." (299-300) is based on "peer reviewed", "knowledge", and "justify" being the most retweeted terms in the evidence category. It's not clear from the manuscript, however, whether these are the most tweeted tweets. There is no comparison to tweets on the same topic which did not contain "common science-based terms" so it's difficult to come to the conclusion that science-based terms don't decrease attention.

2. The tweet IDs and/or their associated tweets do not appear to be provided in the supplementary material.

Additional comments

The manuscript is fairly solid and the application of statistical approaches to this kind of data is well worth examining. With the addition of supporting data, patching up of materials and methods, and a some pulling back of some of the conclusions it makes a decent manuscript. Unfortunately, it does not appear to fall within the scope of the journal which is the only reason for rejection rather than revisions. Good luck.

---

## Round 0.2 · accepted · Accept

The manuscript has been greatly improved compared to the original submission. Although not all the reviewers of the previous version were available to evaluate the resubmission, after carefully reviewing the revised manuscript and the rebuttal letter, I reached the conclusion that all comments were adequately addressed and that the paper is now ready for publication. Congratulations to the authors!

Reviewer 2 ·

Basic reporting

My comments have taken into consideration.

Experimental design

As the authors have refocused the paper, now the manuscript is clearer.

Validity of the findings

My comments have been taken into consideration.